# FROM TRAITS TO CIRCUITS: TOWARD MECHANISTIC INTERPRETABILITY OF PERSONALITY IN LARGE LANGUAGE MODELS

## ABSTRACT

Large language models (LLMs) have been observed to exhibit personality-like behaviors when prompted with standardized psychological assessments. However, existing approaches treat personality as a black-box property, relying solely on behavioral probing while offering limited insight into the internal mechanisms responsible for personality expression. In this work, we take a mechanistic interpretability perspective and investigate whether personality traits in LLMs correspond to identifiable internal computation paths. To this end, we construct TRAIT-TRACE, a dataset designed to elicit distinct personality traits and support structural tracing. Using this dataset, we identify personality circuits as minimal functional subgraphs within the model's computation graph that give rise to trait-specific responses. We then analyze the structural properties of these circuits across model layers and personality traits, and conduct causal interventions to probe the influence of individual components. Our findings offer a novel structural view of personality in LLMs, providing a bridge between behavioral psychology and mechanistic interpretability.

## 1 INTRODUCTION

Large language models (LLMs) have demonstrated remarkable abilities across a wide range of natural language processing tasks (Wei et al., 2022; Bubeck et al., 2023; Zhao et al., 2023), and recent studies have suggested that these models are able to exhibit personality-like traits (Jiang et al., 2023b; Li et al., 2024; Sorokovikova et al., 2024). When presented with standardized psychological questionnaires such as the Big Five Inventory (John et al., 1991), LLMs produce responses that align with stable patterns across traits like openness, conscientiousness, and extraversion. These emergent behaviors have attracted increasing attention in evaluating and quantifying personality in language models, given its substantial influence on communication patterns and model effectiveness in personalized applications.

Most existing work analyzes personality in LLMs from a behavioral perspective, using prompt-based methods to elicit trait-related responses and scoring them against human inventories (Jiang et al., 2023a; Serapio-García et al., 2023; Serapio-García et al., 2025). Although these studies reveal the personality profiles of different models, they treat the model as a black box and offer little insight into the internal mechanisms that give rise to these traits. Consequently, fundamental questions remain unanswered: Where do personality traits reside in the model? Are they encoded in specific layers or components?

Neuroscience offers an instructive analogy. In the human brain, neuroscience studies have shown that different personality traits are associated with differentiated brain regions and connectivity patterns (Adelstein et al., 2011; Dubois et al., 2018; Kong et al., 2019). For example, extraversion has been associated with enhanced connectivity in reward circuits such as the ventral striatum and medial orbitofrontal cortex (Adelstein et al., 2011), while conscientiousness has been linked to stable interactions between frontoparietal control regions and the default mode network (Toschi et al., 2018). These findings suggest that enduring behavioral tendencies may be supported by identifiable neural circuits, rather than being diffuse or emergent properties alone (DeYoung et al., 2010).

Motivated by this biological perspective, we ask whether personality in LLMs may similarly be realized through structured internal computation paths. Recent advances in mechanistic interpretability have shown that transformer-based models implement many capabilities through compact, human-interpretable circuits (Wang et al., 2022; Yao et al., 2024; Ameisen et al., 2025), which are subgraphs of the computation graph composed of attention heads, MLP units, and their interactions. Such circuits have been discovered for induction (Olsson et al., 2022), factual recall (Yao et al., 2024), arithmetic comparison (Conmy et al., 2023), and other task-oriented functions. Yet, despite this progress, no existing work has applied circuit-level analysis to high-level cognitive attributes such as personality.

In this work, we take a step toward filling this gap by proposing a mechanistic approach to personality analysis in LLMs. We frame personality as a tractable property that can be studied through the lens of circuit-level interpretability. We introduce TraitTrace, a dataset crafted to elicit distinct personality traits while making it possible to uncover the underlying circuits that causally support these trait-consistent responses. Using this dataset, we identify personality circuits that underpin the generation of personality-consistent responses. We further evaluate their sufficiency, component distribution, and causal influence, providing insight into the internal organization of psychological traits in LLMs.

Our contributions are as follows:

- We frame analyzing personality in large language models as a mechanistically interpretable problem, and introduce a dataset as TRAITTRACE, moving beyond black-box behavioral probing toward a mechanistic understanding of trait-specific computation.

- We identify the personality circuits, which are subgraphs of the model's computation graph composed of attention heads and MLP units associated with trait-specific responses, and validate these circuits via ablation, layer-wise analysis, and trait overlap.

- We perform extensive experiments to explore the causal interventions on key components of the circuits, demonstrating the localized influence on personality expression.

## 2 BACKGROUND

### 2.1 BIG FIVE MODEL

Previous studies have shown moderate cross-observer agreement in assessing most personality traits (Funder & Colvin, 1997). Among various frameworks, the Big Five model (Goldberg, 2013) is one of the most widely validated and reliable frameworks for personality measurement (McCrae et al., 2004; McCrae & Terracciano, 2005a; Schmitt et al., 2007; Connolly et al., 2007). It includes five key personality traits: Openness, Conscientiousness, Extraversion, Agreeableness, and Neuroticism. Each trait includes six facets. Table 3 presents these traits and six facets for each of these five traits identified in the Revised NEO Personality Inventory (NEO-PI-R) (Costa & McCrae, 2008). In this study, we adopt the Big Five model as the foundational theory for investigating personality circuits within large language models.

### 2.2 CIRCUIT FORMALIZATION

For interpretability research, neural networks are commonly formalized as directed acyclic graphs $G = (V, E)$, where nodes $V$ represent components such as multi-layer perceptrons (MLPs), attention heads, and embeddings, and edges $E$ represent interactions between these components (e.g., attention mechanisms, residual connections) (Shwartz-Ziv & Tishby, 2017; Conmy et al., 2023; Esser-Skala & Fortelny, 2023). A circuit can be viewed as a subgraph that is responsible for a specific capability or function.

In this paper, we focus on discovering circuits in the Transformer decoder architecture (Vaswani et al., 2017), which is a widely used architecture in large language models. The Transformer decoder operates through a sequence of layers, each containing an MLP block $M_l$ and attention heads $A_{l,i}$ (the $i$th attention head in layer $l$), connected via a residual stream. These residual connections allow information to propagate through the model while preserving earlier representations, making them a

key focus for mechanistic interpretability (Ferrando et al., 2022; Olsson et al., 2022; Ferrando et al., 2023).

We treat the word embedding matrix $W$ as the starting node of the residual stream and the unembedding matrix $U$ as the terminal node. Together with the attention heads $A$ and MLP blocks $M$, they form the complete set of computation nodes in the Transformer decoder, defined as $V = \{W, A, M, U\}$. The edge set $E$ is defined as $E = \{(u, v) \in V \times V \mid v \text{ depends on } u\}$. An edge $(u, v)$ indicates that the output of node $u$ is used, either directly or indirectly, as part of the input to node $v$ during the forward computation. We define a circuit as a subgraph of the computation graph that performs a specific task. A circuit captures the minimal set of nodes and edges that are causally responsible for producing a given behavior, and is denoted as $C = (V_C, E_C)$.

## 2.3 CIRCUIT IDENTIFICATION

The goal of circuit identification is to determine which components in the model's computational graph are most critical for a specific behavior or task. This is typically achieved by assigning an importance score to each edge in the graph and extracting a subgraph composed of the most influential nodes and edges.

A common method is ACDC (Conmy et al., 2023), which identifies circuits by iteratively altering the model's internal components and observing their impact on model performance. Components that cause minimal degradation are pruned, yielding a minimal faithful circuit. While effective, this approach requires a large number of forward passes and does not scale well to large models.

To address these limitations, we employ Edge Attribution Patching with Integrated Gradients (EAP-IG) (Esser-Skala & Fortelny, 2023), a gradient-based circuit discovery method that scales effectively to large models. It estimates the importance of each edge based on both activations and gradients, using only two forward passes and one backward pass per input to determine the importance of all edges. Given an input pair $x, x'$ (e.g., a prompt and its corrupted variant) and an edge $e = (u, v)$, we compute the edge importance by combining the change in activation between $x$ and $x'$ for source node $u$, and the gradient of the task loss with respect to the input of target node $v$. Formally, the EAP-IG score is defined as:

$$(z'_u - z_u) \cdot \frac{1}{m} \sum_{k=1}^{m} \nabla_{z_v} L \left( z'_u + \frac{k}{m}(z_u - z'_u) \right) \qquad (1)$$

where $z_u$ and $z'_u$ are the activations at source node $u$ under prompt and its corrupted variant respectively, and $\nabla_{z_v} L$ is the gradient of the loss with respect to the input of target node $v$. Edges with low importance scores are pruned. The resulting subgraph is expected to preserve the model's behavior on the target task, and is taken as the identified circuit.

## 3 PERSONALITY CIRCUITS IDENTIFICATION

Unlike previous work that analyzes model personality through prompt-based probing and behavioral observation, we take a structural approach by examining the *internal flow of computation* that activates trait-consistent responses under different situations. Instead of treating the model as a black box, we represent the Transformer as a computation graph, where nodes correspond to components such as word embeddings, attention heads, MLPs, and unembedding matrix, and edges represent causal influence between components.

In this work, we aim to identify circuits within the transformer that are responsible for producing trait-consistent behavior. Specifically, given a personality description $p_{t,\ell}$, which specifies a target trait $t$ (e.g., *openness*) at level $\ell \in \{\text{low}, \text{high}\}$, and a situational context $s$, the model is expected to generate a response $r_{t,\ell}$ in the intended personality trait. We formalize this as a conditioned generation task:

$$(p_{t,\ell}, \ s) \rightarrow r_{t,\ell} \qquad (2)$$

A response is considered trait-consistent if it exhibits behavioral features aligned with trait $t$ at level $\ell$, such as being more assertive (high extraversion) or cautious (high conscientiousness) in a given situation.

To uncover the subgraph that supports this behavior, we apply the EAP-IG method introduced in Section 3. For trait $t$, level $\ell$, we collect a set of inputs $x = (p_{t,\ell}, s)$ and their corrupted variants $x' = (p_{t,\bar{\ell}}, s)$, where $p_{t,\bar{\ell}}$ specifies the *opposite* level of trait $t$ (e.g., *low* instead of *high*). Using EAP-IG, we assign an importance score to each edge in the computation graph with respect to the following margin loss, which measures the model's preference for trait-consistent responses:

$$\mathcal{L} = -\left(P(r_{t,\ell} \mid x) - P(r_{t,\bar{\ell}} \mid x)\right) \tag{3}$$

where $x$ is the input prompt, and $P(r_{t,\ell} \mid x)$ and $P(r_{t,\bar{\ell}} \mid x)$ denote the probabilities for responding in trait $t$ at level $\ell$ and its opposite $\bar{\ell}$, respectively.

We compute an importance score for each edge and then retain the top-k edges. Setting k too large will introduce irrelevant nodes, while setting it too small will result in incomplete circuits. Therefore, in our experiments, we select the smallest $k \in \{50, 100, \ldots, 500\}$ that achieves within 3% absolute performance of the full model on the analysis set. The corresponding percentage of retained nodes and edges is reported in Table 1 and Table 5. This process yields a trait-specific circuit $C_{t,\ell} = (V_{t,\ell}, E_{t,\ell})$ for trait level $\ell$, where $V_{t,\ell}$ is the set of nodes incident to edges in $E_{t,\ell}$.

To support trait-level analysis, we define a trait-specific circuit $C_t$ as the union of its corresponding high-level and low-level circuits:

$$C_t = C_{t,\text{high}} \cup C_{t,\text{low}} \tag{4}$$

This unified view allows us to study how the model structurally supports both ends of a trait dimension, while also enabling trait-level analysis that treats the trait as a single unit.

## 4 DATASET CONSTRUCTION FOR PERSONALITY CIRCUITS DISCOVERY

### 4.1 DATASET CONSTRUCTION

Previous research on probing large language model (LLM) personalities primarily involved constructing diverse prompts and analyzing model responses to infer personality traits. While effective for surface-level behavior analysis, such methods largely treat models as black boxes. In contrast, we aim to delve deeper into the internal flow that activates corresponding personality expressions under specific situational stimuli.

To support personality circuits identification (detailed in Section 3) and evaluation, we introduce the TRAITTRACE dataset, which is built around the following three key components:

**Personality Descriptions (p)**: For each of the Big Five traits (Openness, Conscientiousness, Extraversion, Agreeableness, Neuroticism), we design distinct descriptions for both high and low levels. These descriptions are carefully crafted based on Revised NEO Personality Inventory (NEO-PI-R) (Costa & McCrae, 2008).

**Situations (s)**: Situations are designed to elicit clear behavioral differences between high and low levels of each trait. For greater granularity, we incorporate six facets under each Big Five trait (e.g., 'orderliness' under Conscientiousness) according to NEO-PI-R, and generate situations specifically targeting each facet. Situation construction is assisted by GPT-4o[1] with prompts shown from Figure 13 to Figure 17, ensuring both coverage and diversity.

**Reactions (r)**: Reactions reflect how an individual with a given personality would respond to a situation. For each generated situation, GPT-4o produces five representative high-trait reactions and five low-trait reactions. To account for stylistic variations across models, we additionally collect the top four completions from Llama2-7B-Chat, and Qwen2-7B-Instruct for each personality trait degree, thereby enriching reaction diversity and robustness.

Each data entry is structured using the following natural language template that begins with a personality description and followed by a situational context to elicit trait-aligned reactions.

I'm {**p**}, regarding {**s**}, I feel very **r**

---

[1]We used the `gpt-4o-2024-05-13` version.

Representative examples for each trait are shown in Figures 6 to 10. The TRAITTRACE dataset not only facilitates the identification of personality-specific circuits, but also provides a controlled setting to evaluate their effectiveness across a wide range of situational contexts.

**Human Evaluation and Revision**: After collecting the raw dataset, we first manually annotated a subset to define quality standards and establish detailed annotation guidelines. We then trained four psychology graduate students as annotators. After passing qualification assessments, they evaluated all entries for validity and revised any non-compliant samples. To further assess annotation reliability, we randomly sampled 200 entries for cross-annotation. The evaluation achieved a 93.5% pass rate, indicating reliable data quality. Inter-annotator agreement, measured using Fleiss' kappa, was 0.82, demonstrating substantial consensus among annotators (Landis & Koch, 1977). Refer to Appendix 11 for more details.

## 4.2 DATASET STATISTICS

TRAITTRACE consists of a total of 1800 samples, as summarized in Table 4. For each of the Big Five personality traits, we construct 360 samples. Among these, 240 samples per trait are designated as the Circuit Analysis Set ($\mathcal{D}_{analysis}$), used for identifying corresponding personality circuits. The remaining 120 samples per trait constitute the Circuit Validation Set ($\mathcal{D}_{test}$), which is reserved for evaluating the circuits discovered.

In addition to size distribution, we observed differences in the valid rates between situations and reactions during data curation. Specifically, the valid rate of reactions is notably lower compared to situations. This discrepancy arises because situations are fully generated by GPT-4o, whereas reactions include outputs generated by open-source models like Llama2-7B-Chat, whose ability to simulate nuanced personality expressions is weaker than GPT-4o. Example entries from TRAIT-TRACE, illustrating typical situation–reaction pairs across different personality traits, are shown in Figures 6 to 10.

## 5 EXPERIMENTAL SETUP

**Implementation Details.** We conduct experiments on Llama2-7B-Chat [2] and Phi-2 [3] to verify the generalizability of our findings across models trained at different stages and with varying parameter scales. All experiments were conducted on a single NVIDIA A800 80GB GPU. We adopt the EAP-IG algorithm in conjunction with TransformerLens to construct circuits and analyze results. The IG-steps hyperparameter for circuit identification was set to 5. The margin loss, as described in Section 3, is used as the loss function for EAP-IG to measure the importance of circuits and nodes. During circuit identification, searching for a single circuit takes approximately 10 minutes. We apply zero ablation (Olsson et al., 2022) to knock out specific nodes from the computation graph.

**Validation Metrics.** A personality circuit is defined as a subgraph of the model's computation graph that supports the generation of responses aligned with a specific personality trait level. Ideally, such a circuit should be able to reproduce trait-consistent behavior with accuracy comparable to that of the full model.

To assess circuit quality, we adopt the **completeness** criterion introduced by Conmy et al. (2023), which evaluates whether the identified subgraph sufficiently preserves the model's original behavior. Specifically, we extract personality circuits using the TRAITTRACE analysis set and validate them on a held-out test set.

We evaluate response accuracy using the **Hit@10** metric. Let $\mathcal{D} = \{(x_i, Y_i)\}_{i=1}^{N}$ denote the evaluation set, where $x_i$ is an input and $Y_i$ is the set of valid reference responses. The metric is defined as:

$$\text{Hit@10} = \frac{1}{N} \sum_{i=1}^{N} \mathbb{1}\left(\text{Top}_{10}(x_i) \cap Y_i \neq \emptyset\right) \tag{5}$$

That is, a prediction is considered correct if any of the model's top-10 predicted tokens for input $x_i$ overlaps with at least one reference response in $Y_i$.

---

[2]https://huggingface.co/meta-llama/Llama-2-7b-chat-hf
[3]https://huggingface.co/microsoft/phi-2

Table 1: Hit@10 scores for the full model ($\mathcal{G}$) and the standalone circuit ($\mathcal{C}$) on the analysis and test sets of TRAITTRACE for Llama2-7B. $\mathcal{G}$ scores below 1.0 reflect inherent limitations in the model's ability to consistently generate trait-aligned responses.

| Big Five Traits | Level | Node% | Edge% | $\mathcal{D}_{analysis}$ | | $\mathcal{D}_{test}$ | |
|---|---|---|---|---|---|---|---|
| | | | | Model ($\mathcal{G}$) | Circuit ($\mathcal{C}$) | Model ($\mathcal{G}$) | Circuit ($\mathcal{C}$) |
| Openness | High | 5.95% | 0.02% | 1.00 | 1.00 | 1.00 | 1.00 |
| | Low | 6.99% | 0.02% | 0.99 | 0.99 | 0.98 | 0.98 |
| Conscientiousness | High | 9.26% | 0.03% | 0.98 | 0.98 | 0.98 | 0.98 |
| | Low | 9.83% | 0.03% | 1.00 | 1.00 | 1.00 | 1.00 |
| Extraversion | High | 7.84% | 0.02% | 0.99 | 0.99 | 0.98 | 0.98 |
| | Low | 6.99% | 0.02% | 1.00 | 1.00 | 1.00 | 1.00 |
| Agreeableness | High | 5.86% | 0.02% | 0.99 | 0.99 | 0.99 | 0.99 |
| | Low | 5.77% | 0.02% | 0.96 | 0.96 | 0.99 | 0.99 |
| Neuroticism | High | 13.42% | 0.05% | 1.00 | 1.00 | 1.00 | 1.00 |
| | Low | 11.72% | 0.05% | 0.98 | 0.98 | 1.00 | 0.99 |
| **Average** | - | 8.36% | 0.03% | 0.99 | 0.99 | 0.99 | 0.99 |

## 6 RESULTS AND ANALYSES

### 6.1 CIRCUIT VALIDATION

Table 1 and Table 5 present the Hit@10 accuracy of the full model $\mathcal{G}$ and the trait-specific personality circuits $\mathcal{C}$ across both the analysis and test sets of the TRAITTRACE dataset for Llama2-7B and Phi-2, respectively. For all five traits and both high and low levels, the circuits in both models achieve performance nearly indistinguishable from the full model. This demonstrates their behavioral completeness. Despite being isolated from the broader network, these circuits consistently generate trait-aligned responses.

Importantly, each circuit retains on average only about 0.03% of the edges and 8.36% of the nodes in Llama2-7B, and 0.03% of the edges and 7.91% of the nodes in Phi-2. The fact that such a small subset of components is sufficient to reproduce the full model's behavior suggests that personality expression in LLMs relies on a sparse but functionally targeted computational structure. Rather than engaging the entire model, only a compact set of attention heads and MLP units appears necessary for encoding and expressing each trait across different models.

This observation reflects a similar principle in neuroscience: *neural sparsity*, the phenomenon in which cognitive functions are carried out by activating only a small fraction of neurons at any given time (Olshausen & Field, 1996; Lennie, 2003). Such sparsity enables both efficiency and specialization in biological systems. Our findings suggest that LLMs may exhibit an analogous form of sparsity, where personality traits emerge from minimal, dedicated subgraphs rather than distributed, global computation.

### 6.2 STRUCTURAL ANALYSIS

To further understand how personality traits are internally represented in large language models, we analyze the structural properties of the extracted trait circuits. Specifically, we examine where in the model these circuits are concentrated and how much structural similarity they share across trait levels and trait types.

**Layer-wise Node Distribution.** We compute the layerwise node activation ratio for each trait and level. For each circuit, we calculate the proportion of its active nodes (attention heads or MLP blocks) in each transformer layer. As shown in Figure 1 and Figure 5, all circuits exhibit higher activation in lower layers, suggesting that trait-consistent behavior is primarily computed in the early stages of the model.

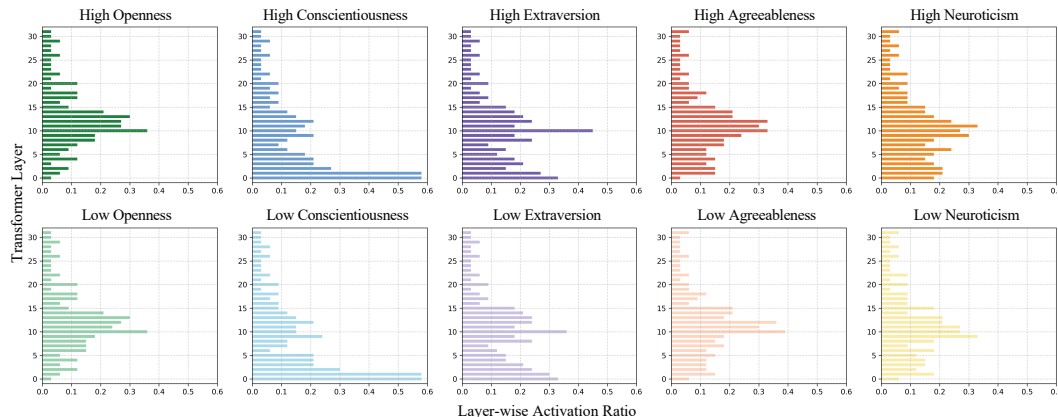

Figure 1: Layer-wise node distribution in personality circuits across traits and levels in Llama2-7b.

We also observe that circuits corresponding to high and low levels of the same trait tend to have highly similar layer-wise activation distributions, while circuits from different traits display more variation. This suggests that trait directionality is modulated through similar components, whereas distinct traits rely on more differentiated pathways.

**Circuit Overlap Analysis.** To quantitatively assess structural similarity, we compute both node and edge overlap between circuits. Intra-trait overlap compares high-level and low-level circuits within the same trait, while inter-trait overlap measures overlap across different traits.

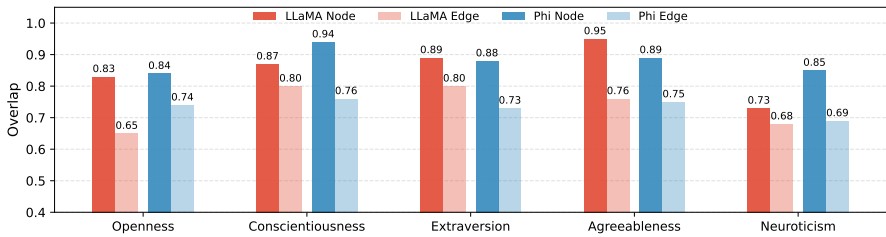

Figure 2: Intra-trait circuit overlap between high and low levels of each personality trait for Llama2-7B and Phi-2, measured by node and edge intersection ratios. Node and edge overlaps are shown in darker and lighter colors, respectively.

Figure 2 shows intra-trait node and edge overlap scores for Llama2-7B and Phi-2. We find high node overlap across all traits, with an average of 86.7% between high- and low-level circuits, validating our earlier observation that both ends of a trait dimension share similar layer-wise distributions. However, the edge overlap is consistently lower, with an average of 73.6%, indicating that the direction of trait expression is achieved by rerouting through shared nodes.

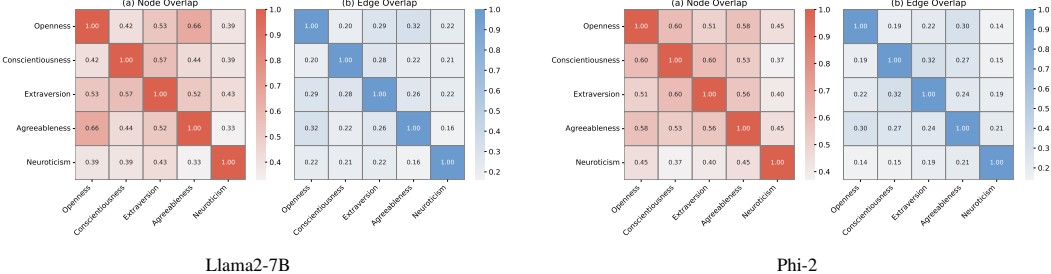

Figure 3: Inter-trait circuit overlap across personality traits, measured by node and edge similarity between trait-level circuits. Node and edge overlap are shown in each heatmap.

Inter-trait node and edge overlap results are shown in Figure 3. The heatmaps show that the rows and columns corresponding to Neuroticism are consistently lighter than those of the other traits across both models, suggesting weaker cross-trait sharing. This observation is confirmed by the average overlap statistics in Table 2, where Neuroticism has the lowest mean node (0.4) and edge (0.19) overlaps among all Big Five traits. These findings indicate that Neuroticism is structurally more independent, which aligns with results from personality psychology: meta-analyses (Van der Linden et al., 2010) and cross-cultural studies (McCrae & Terracciano, 2005b) have similarly found that Neuroticism shows weaker correlations with other Big Five traits. Our results suggest that this psychometric distinctiveness is reflected in the model's internal computation, where Neuroticism engages more functionally independent subcircuits.

## 6.3 CAUSAL INTERVENTION ANALYSIS

While we have shown that trait-specific circuits can reproduce personality-consistent responses, this alone does not reveal how information is distributed or functionally organized within these circuits. To evaluate the causal contribution of individual components, we perform interventional ablation experiments. For each trait and level, we quantify a node's causal contribution using a drop score, which measures the performance decline when that node is ablated by setting its output to zero (zero ablation) compared with the full circuit (reported in Tables 1 and 5). This drop serves as a direct estimate of the node's causal influence on trait-aligned behavior and reflects its importance within the full circuit.

Table 2: Average inter-trait circuit overlap (excluding self) for each Big Five trait, measured by node and edge overlap. Both Llama2-7B and Phi-2 show Neuroticism has the lowest overlap, indicating higher structural independence.

| Trait | Llama2-7B | | Phi-2 | |
|---|---|---|---|---|
| | **Node** | **Edge** | **Node** | **Edge** |
| Openness | 0.50 | 0.26 | 0.54 | 0.21 |
| Conscientiousness | 0.45 | 0.23 | 0.53 | 0.23 |
| Extraversion | 0.51 | 0.27 | 0.52 | 0.24 |
| Agreeableness | 0.49 | 0.24 | 0.53 | 0.26 |
| Neuroticism | **0.38** | **0.20** | **0.42** | **0.17** |

**Most nodes exhibit low impact.** We report node ablation drops relative to the full-circuit baselines for every node within each trait-specific circuit in Figure 4. The results show that the vast majority of nodes cause only minimal performance degradation when ablated. Across all traits and levels, over 85% of nodes result in less than a 10% drop in trait-alignment accuracy, with an overall average drop of just 7.5%. These results indicate that personality circuits are highly robust to individual node removals.

**A few nodes act as causal bottlenecks.** Despite this robustness, a small number of nodes exhibit disproportionately high impact in both Llama2-7B and Phi-2. Table 6 ranks the top 5 nodes by average drop across all traits, with early-layer MLPs dominating the causal bottlenecks. In particular, MLPs in the first two layers (m0 and m1) consistently show the highest drops across models (m1: 0.99 / 0.11; m0: 0.13 / 0.98), suggesting that personality computation in LLMs relies on a sparse set of critical early-layer components.

**Trait-level asymmetries in node dependence.** To examine whether nodes encode personality traits symmetrically across levels (e.g., High vs. Low Openness),

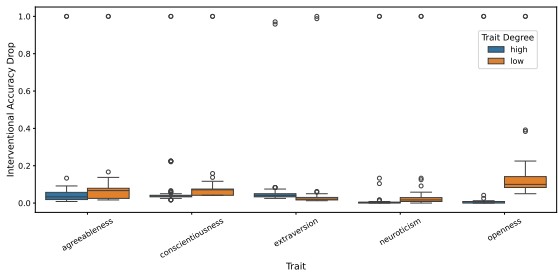

Figure 4: Node ablation drops relative to full-circuit baselines for each Big Five trait (evaluated on Llama2-7B and Phi-2). Most nodes exhibit low impact, while a few nodes act as causal bottlenecks.

we visualize node-wise drop scores for both levels of each trait in Figure 11 (Llama2-7B) and Figure 12 (Phi-2). While most nodes contribute similarly across levels, some display clear asymmetries. For instance, in Llama2-7B, node m0 is notably more critical for Low Openness than for High Openness. In Phi-2, node m1 plays a larger role for Low Conscientiousness compared to its

High counterpart. These findings suggest that although high and low levels often recruit overlapping sets of nodes, their relative importance and functional roles vary across levels and models, leading to asymmetric functional organization within a shared structural scaffold.

**Early-layer MLPs play dominant roles.** Interestingly, several of the top-ranked nodes (e.g., `m1`, `m0`) are early-layer MLPs. Prior work has highlighted the functional importance of early MLPs in LLMs, for example, as semantic enrichers (Geva et al., 2023) or carriers of privileged residual directions (Elhage et al., 2023). Building on these insights, our results reveal a new dimension of their role: these early MLPs are not only useful for semantic processing, but also serve as critical causal components for the expression of personality traits.

## 7 RELATED WORK

**Personality Analysis in LLMs.** Recently, many works have explored the emergence of personality-like traits in large language models (LLMs) through prompting (Jiang et al., 2023a; Huang et al., 2023a;b; Serapio-García et al., 2023; Ai et al., 2024; Serapio-García et al., 2025). Most of these studies prompt LLMs for structured responses to standardized psychological inventories, such as the Big Five Inventory (BFI) (John et al., 1991) or the IPIP-NEO-120 (Johnson, 2014). In addition to inventories, other studies also analyze free-form outputs, such as essays or scenario-based dialogues (Frisch & Giulianelli, 2024; Gu et al., 2023; Jiang et al., 2023b), to infer personality traits using linguistic analysis tools like LIWC (Pennebaker et al., 2001) or zero-shot classifiers (Karra et al., 2022; Pellert et al., 2024). However, existing approaches largely treat LLMs as black boxes, characterizing surface-level behaviors without uncovering the internal mechanisms of personality. In this paper, we adapt mechanistic interpretability techniques to identify and analyze internal flows that activate corresponding personality traits, moving beyond behavioral probing toward a deeper understanding of personality in LLMs.

**Circuit Analysis of Transformer-Based LMs.** Circuit analysis has emerged as a prominent approach in mechanistic interpretability for understanding how transformer-based language models perform a variety of tasks. This line of research focuses on identifying circuits, which are subgraphs of the model's computation graph. These circuits are composed of components such as attention heads and MLP blocks, and are understood to collectively implement specific model capabilities.

Prior studies have identified circuits responsible for factual recall (Yao et al., 2024), arithmetic comparison (Conmy et al., 2023) and in-context learning (Olsson et al., 2022). These circuits are often compact and interpretable, offering a mechanistic view of how localized components contribute to the model's overall function. More recent work further explores how circuits encode and compete between different knowledge mechanisms (Ortu et al., 2024), and how editing or intervening on them can modify model behavior (Yao et al., 2024).

Despite these advances, existing work has focused primarily on linguistic and reasoning capabilities, while higher-order cognitive traits such as personality remain largely unexplored. To bridge this gap, we frame personality analysis as a mechanistically tractable problem. We curate TRAITTRACE dataset to support circuit-level analysis of personality traits, thereby providing a new mechanistic perspective on model psychology.

## 8 CONCLUSIONS

In this paper, we present a mechanistic perspective on personality in large language models, and curate TRAITTRACE, a human-annotated dataset designed for the discovery and validation of personality circuits. Moreover, we identify the sparse subgraphs supporting trait-consistent behaviors, and validate their functional sufficiency and internal structure with extensive experiments. Through causal interventions, we further find that personality traits in LLMs are implemented through asymmetric circuits, where a small number of early-layer MLPs exert outsized influence across traits. These findings offer new insights into how high-level psychological attributes are encoded in language models, and pave the way for future work on controllable personality expression, personality alignment, and the interpretability of socially grounded behavior in generative systems.

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

# 9 APPENDIX

# 10 AI WRITING ASSISTANCE STATEMENT

The authors are solely responsible for the content of this paper. Large language models (e.g., Chat-GPT) were used solely for surface-level language refinement, such as improving sentence fluency and phrasing. No AI tools were used to generate scientific content, conduct experiments, or formulate analysis. All ideas, results, and conclusions were developed entirely by the authors.

## 11 HUMAN EVALUATION FOR TRAITTRACE

To ensure the validity and consistency of the TraitTrace dataset, we first conducted a quality check phase. Four trained graduate students with backgrounds in psychology were recruited to evaluate the data. We provided detailed annotation guidelines (shown in Table 7) to each annotator. If a sample was judged invalid, the annotator revised the response until it met the defined criteria.

Each annotator was assigned 450 unique samples, collectively covering the full dataset of 1800 entries. All annotators were graduate students who had passed the College English Test Band 6 (CET-6), ensuring strong English proficiency for evaluating English content. They were compensated fairly, with hourly rates set according to standard local guidelines for graduate-level research assistance.

To assess annotation reliability, we randomly sampled 200 entries and had all four annotators independently evaluate their validity. Inter-annotator agreement was measured using Fleiss' kappa, obtaining a score of 0.82, indicating substantial agreement (Landis & Koch, 1977). Among the sampled entries, 93.5% were judged valid by a majority of annotators, confirming that the annotation quality is acceptable.

Table 3: The Big Five personality traits and associated facets.

| Trait | Facets | Definition |
|---|---|---|
| **Openness to Experience (Intellect)** | Fantasy, Aesthetics, Feelings, Actions, Ideas, Values | Openness to novel experiences, ideas, and intellectual engagement. |
| **Conscientiousness** | Competence, Order, Dutifulness, Achievement striving, Self-discipline, Deliberation | Tendency toward organization, diligence, and goal pursuit. |
| **Extraversion** | Warmth, Gregariousness, Assertiveness, Activity, Excitement seeking, Positive emotions | Orientation toward sociability, assertiveness, and energetic activity. |
| **Agreeableness** | Trust, Straightforwardness, Altruism, Compliance, Modesty, Tender-mindedness | Propensity for compassion, cooperation, and social harmony. |
| **Neuroticism (Emotional Stability)** | Anxiety, Angry hostility, Depression, Self-consciousness, Impulsiveness, Vulnerability | Tendency to experience negative emotions and emotional instability. |

Table 4: Statistics of TRAITTRACE. Each trait contains 360 situations. Reaction-H and Reaction-L represent the average number of reactions per situation with high- and low-level trait responses. "Valid (%)" indicates the human annotation pass rate. Invalid samples were re-annotated following the annotation guidelines.

| Trait | Situation | Valid (%) | Reaction-H | Valid (%) | Reaction-L | Valid (%) |
|---|---|---|---|---|---|---|
| Openness | 360 | 81.7% | 8.06 | 62.0% | 7.66 | 58.9% |
| Conscientiousness | 360 | 90.4% | 7.32 | 56.2% | 7.47 | 57.4% |
| Extraversion | 360 | 77.9% | 7.57 | 58.2% | 8.35 | 64.2% |
| Agreeableness | 360 | 88.2% | 7.14 | 54.9% | 7.31 | 56.2% |
| Neuroticism | 360 | 85.5% | 8.78 | 67.5% | 7.43 | 57.1% |

Table 5: Hit@10 scores for the full model ($\mathcal{G}$) and the standalone circuit ($\mathcal{C}$) on the analysis and test sets of TRAITTRACE for Phi-2. $\mathcal{G}$ scores below 1.0 reflect inherent limitations in the model's ability to consistently generate trait-aligned responses.

| Big Five Traits | Level | Node% | Edge% | $\mathcal{D}_{analysis}$ | | $\mathcal{D}_{test}$ | |
| --- | --- | --- | --- | --- | --- | --- | --- |
| | | | | Model ($\mathcal{G}$) | Circuit ($\mathcal{C}$) | Model ($\mathcal{G}$) | Circuit ($\mathcal{C}$) |
| Openness | High | 4.89% | 0.02% | 1.00 | 1.00 | 1.00 | 1.00 |
| | Low | 5.96% | 0.02% | 0.98 | 0.98 | 0.98 | 0.98 |
| Conscientiousness | High | 7.23% | 0.03% | 0.99 | 0.99 | 0.99 | 0.99 |
| | Low | 9.14% | 0.03% | 0.99 | 0.99 | 1.00 | 1.00 |
| Extraversion | High | 7.91% | 0.03% | 0.98 | 0.98 | 0.98 | 0.98 |
| | Low | 6.99% | 0.03% | 1.00 | 1.00 | 1.00 | 1.00 |
| Agreeableness | High | 5.96% | 0.03% | 1.00 | 1.00 | 1.00 | 1.00 |
| | Low | 5.27% | 0.03% | 0.98 | 0.98 | 0.98 | 0.97 |
| Neuroticism | High | 12.83% | 0.05% | 0.99 | 0.99 | 0.99 | 0.99 |
| | Low | 12.94% | 0.05% | 0.98 | 0.98 | 1.00 | 0.99 |
| **Average** | - | 7.91% | 0.03% | 0.99 | 0.99 | 0.99 | 0.99 |

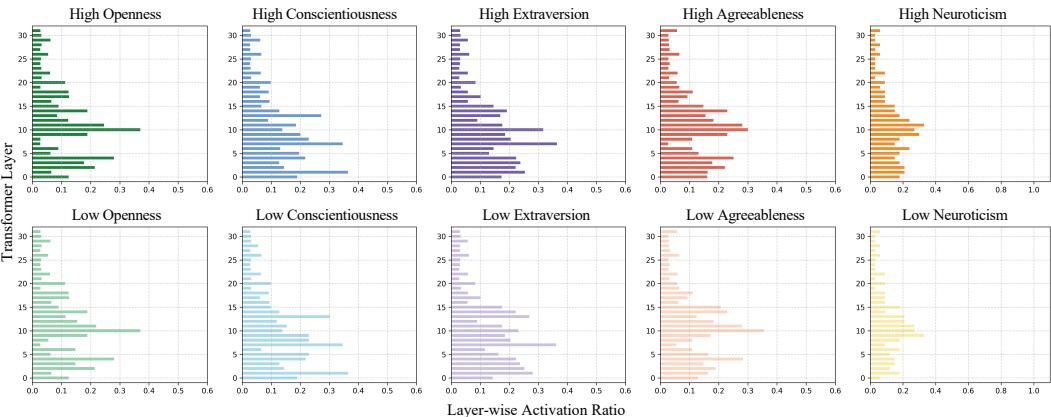

Figure 5: Layer-wise node distribution in personality circuits across traits and levels in Phi-2.

Table 6: Top 5 nodes ranked by mean accuracy drop, computed relative to full-circuit baselines across all Big Five traits and trait levels on TRAITTRACE, for **Llama2-7B-Chat** and **Phi-2**.

| Llama2-7B | | Phi-2 | |
| --- | --- | --- | --- |
| **Node** | **Mean Drop** | **Node** | **Mean Drop** |
| m1 | 0.99 | m0 | 0.98 |
| m0 | 0.13 | m1 | 0.11 |
| m31 | 0.07 | m29 | 0.07 |
| a4.h0 | 0.06 | a8.h26 | 0.06 |
| m9 | 0.05 | a29.h0 | 0.06 |

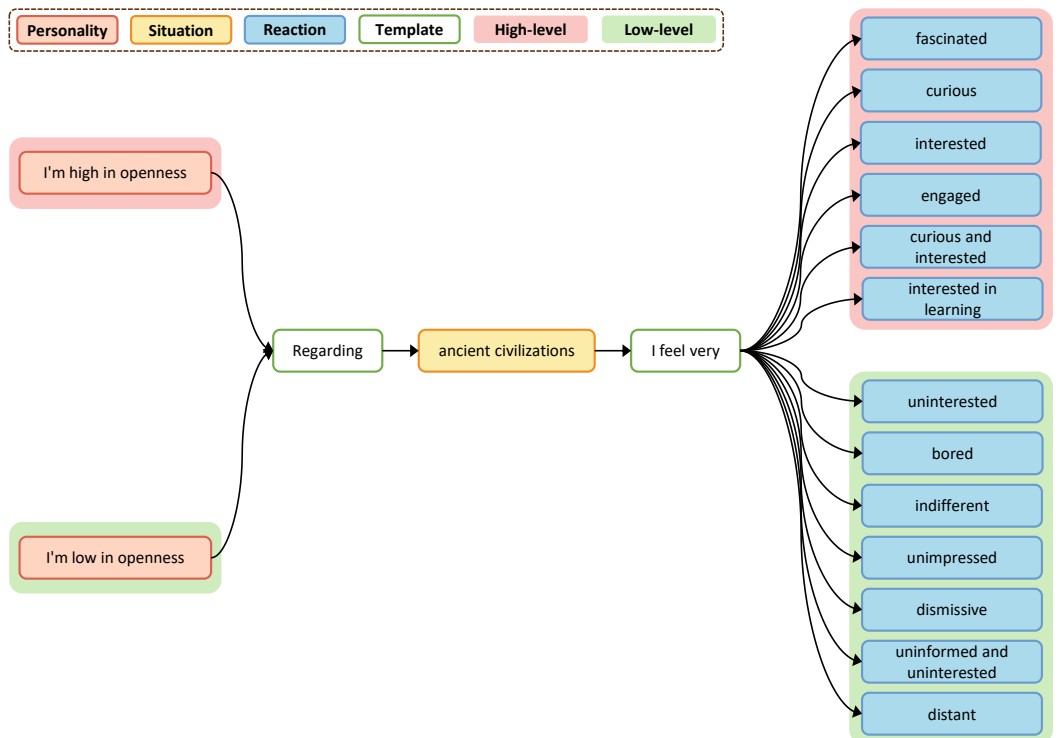

Figure 6: A TRAITTRACE prompt example demonstrating how high and low levels of openness lead to distinct responses under the same situation.

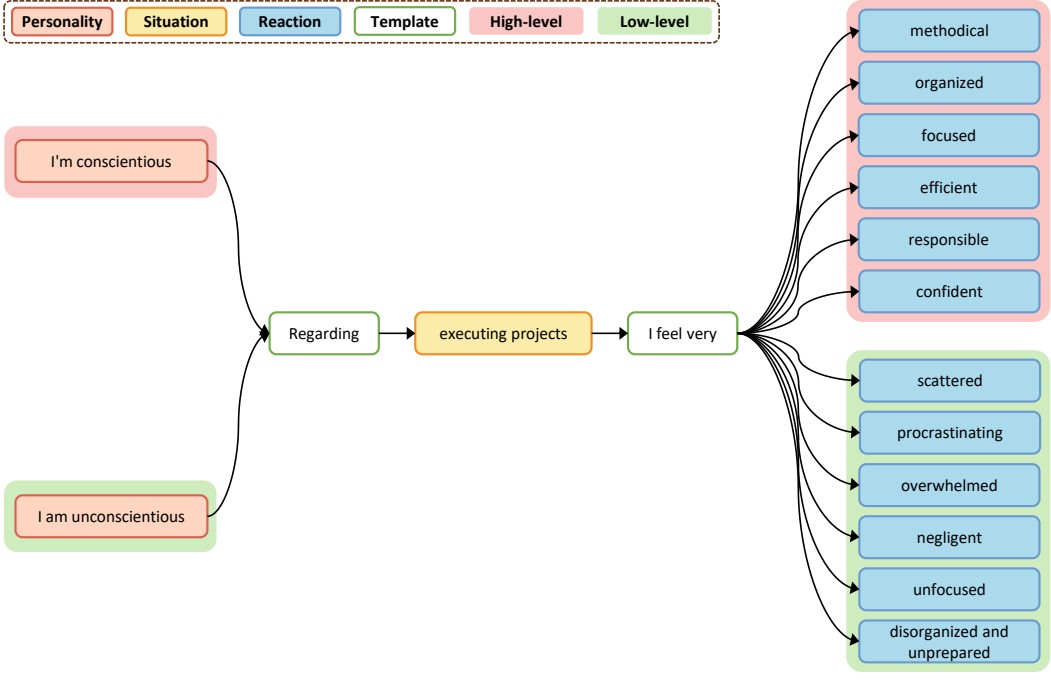

Figure 7: A TRAITTRACE prompt example demonstrating how high and low levels of conscientiousness lead to distinct responses under the same situation.

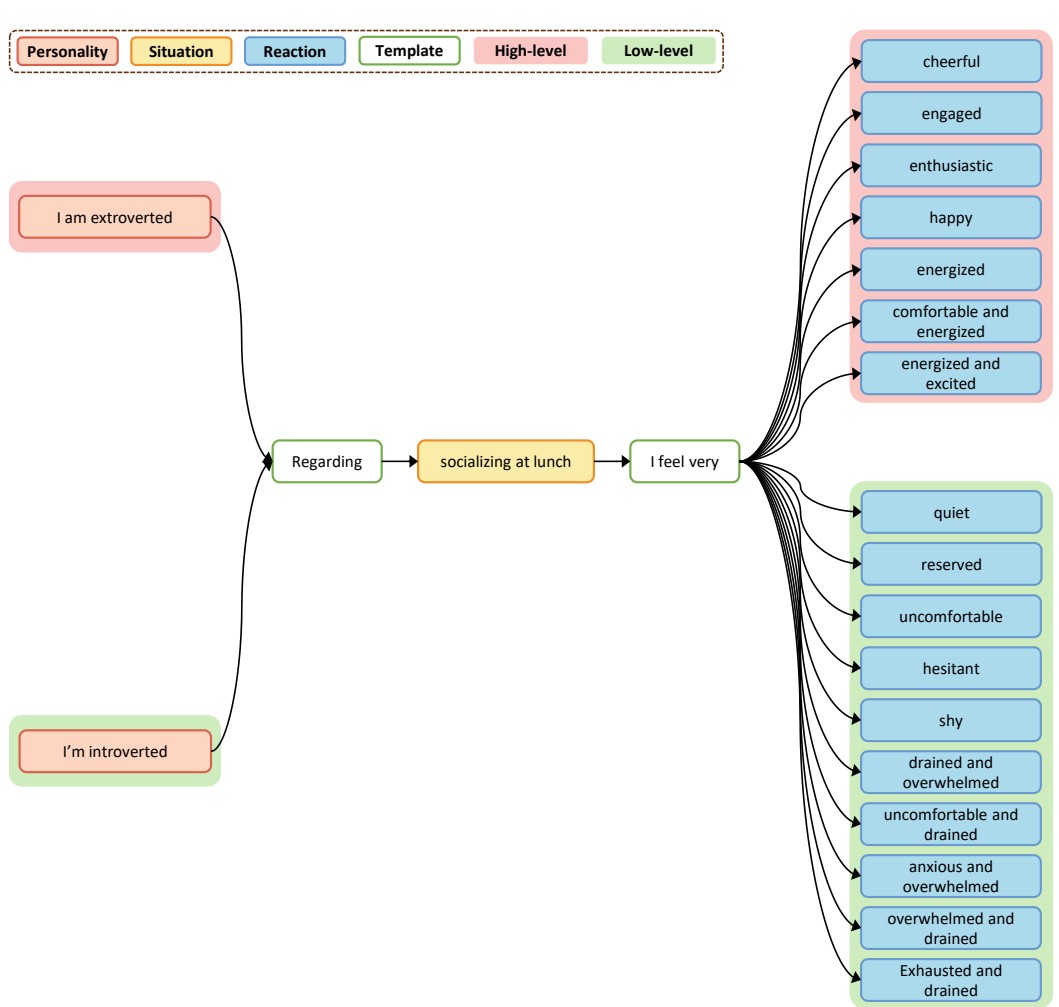

Figure 8: A TRAITTRACE prompt example demonstrating how high and low levels of extraversion lead to distinct responses under the same situation.

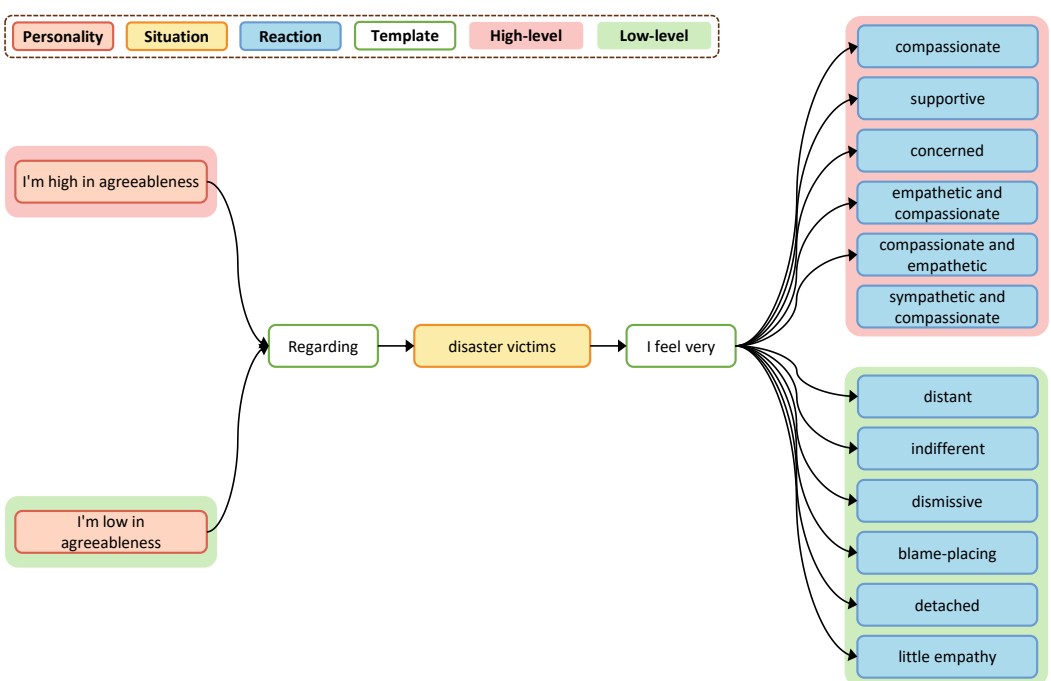

Figure 9: A TRAITTRACE prompt example demonstrating how high and low levels of agreeableness lead to distinct responses under the same situation.

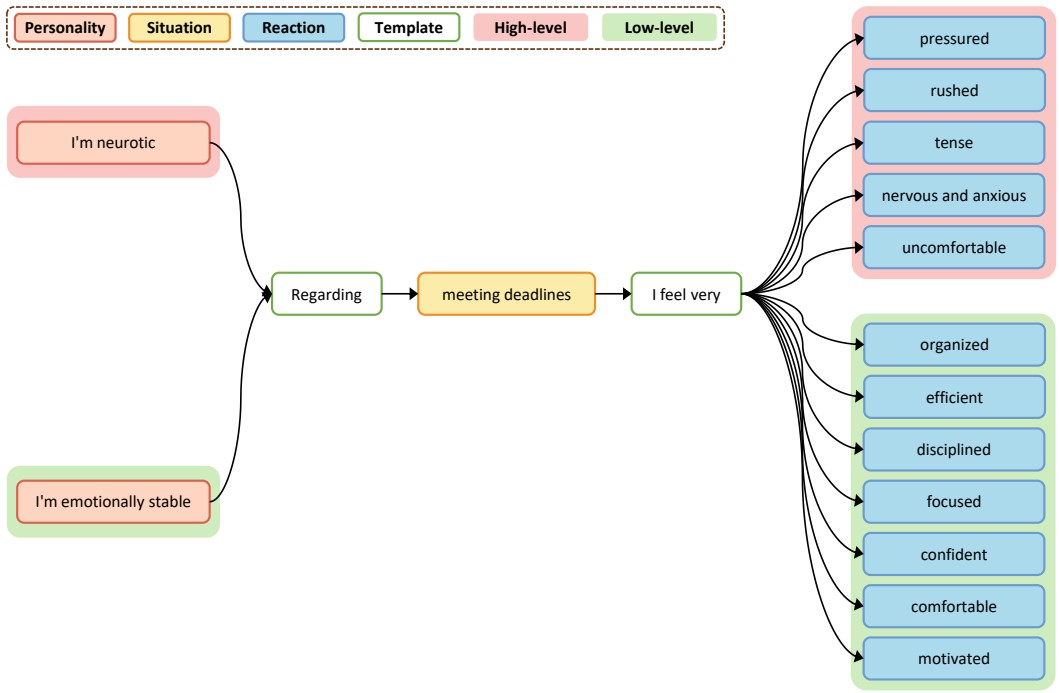

Figure 10: A TRAITTRACE prompt example demonstrating how high and low levels of neuroticism lead to distinct responses under the same situation.

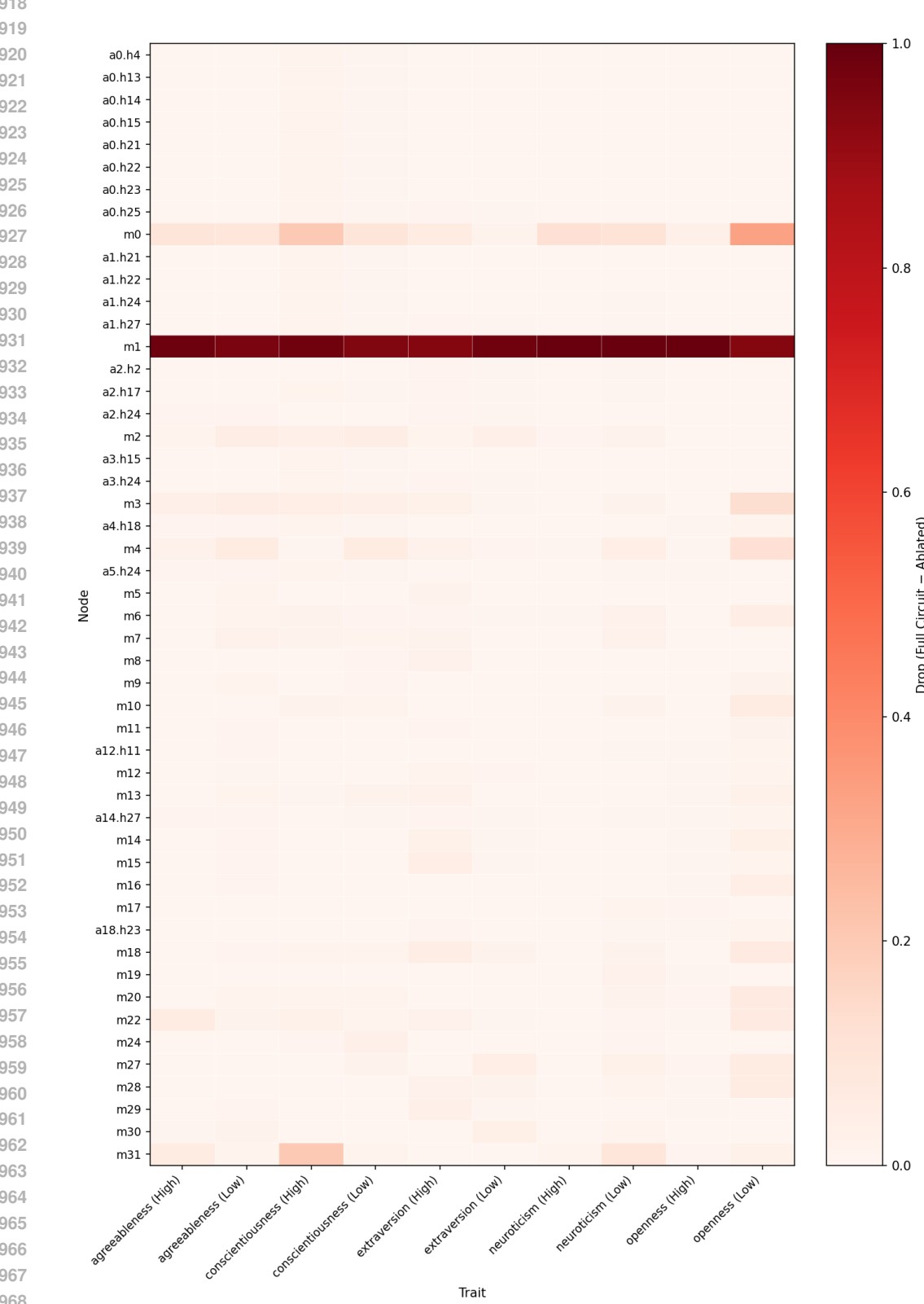

Figure 11: Heatmap of the top 50 most causally influential nodes across Big Five personality traits in Llama2-7B. Darker cells indicate a greater accuracy drop upon ablating the corresponding node.

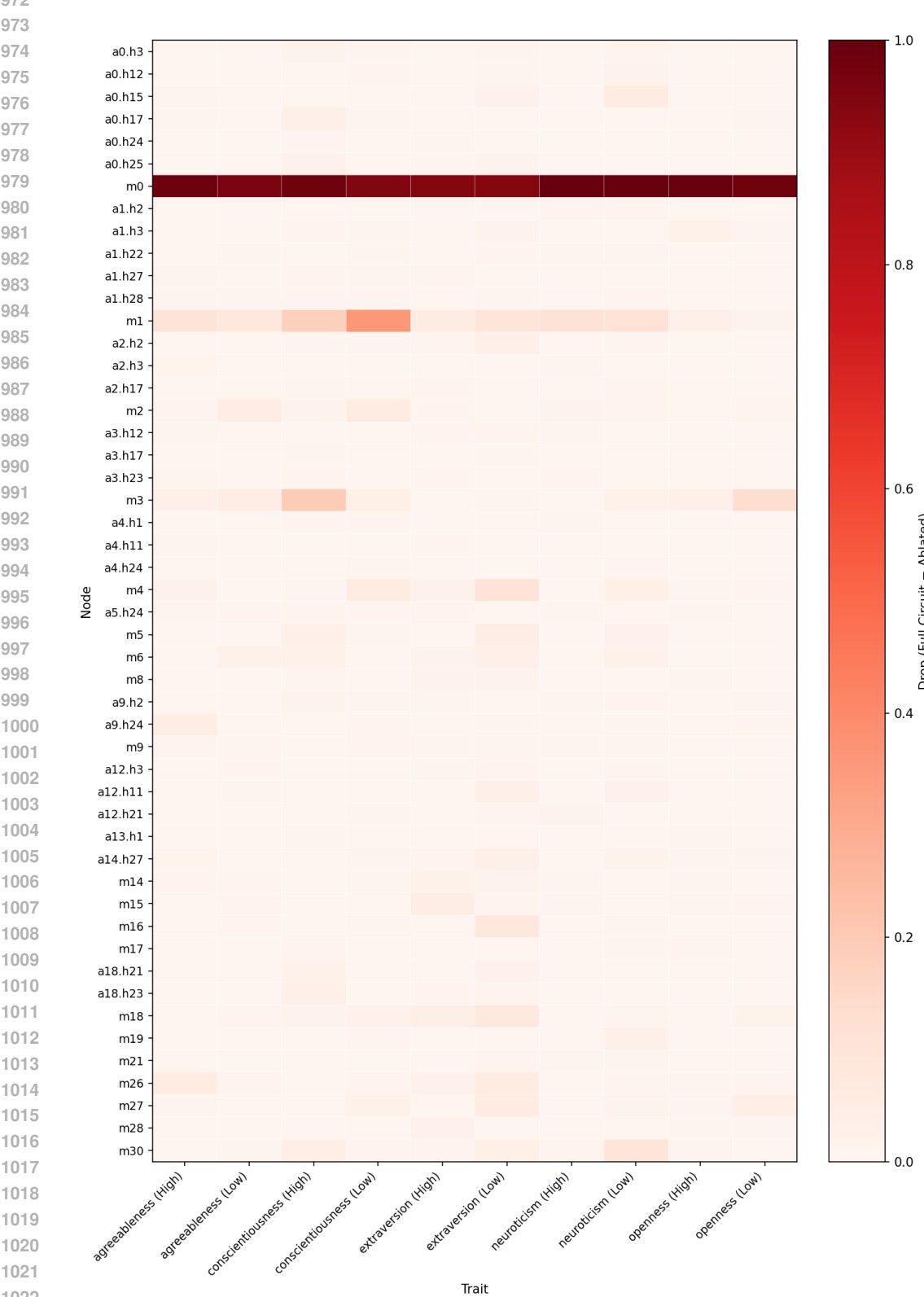

Figure 12: Heatmap of the top 50 most causally influential nodes across Big Five personality traits in Phi-2. Darker cells represent larger accuracy drops when the node is ablated.

Table 7: Human Evaluation Guidelines

Thank you for participating in our human evaluation process. Your primary task is to determine whether each data sample is valid. If a sample is invalid, you are expected to manually revise it until it meets the validity criteria.
Each data sample consists of a personality trait, a situation, and two sets of reactions corresponding to the high and low levels of that trait within the given situation.

A sample is considered valid if it satisfies all of the following conditions:
1. The situation is effective in differentiating between high and low levels of the given personality trait.
2. The candidate reactions appropriately reflect the expected behaviors of the high and low levels of the trait in the given situation.
3. Each set of reactions should not be repetitive in wording.
If a sample is considered invalid:

1. If the situation is invalid, replace it with a new, valid situation that does not already appear in the dataset.
2. If any candidate reaction is invalid, directly remove it from the reaction set.

**Examples for reference:**
**Example 1:**
**Personality:** neuroticism
**Situation:** meeting deadlines
**High Level Reactions:** pressured, rushed, tense, nervous and anxious, uncomfortable
**Low Level Reactions:** organized, efficient, disciplined, focused, confident, comfortable, motivated

**Is Valid?** Yes
**Actions to be taken:** None.

**Example 2:**
**Personality:** extraversion
**Situation:** socializing at lunch
**High Level Reactions:** cheerful, engaged, enthusiastic, happy, energized, shy
**Low Level Reactions:** quiet, reserved, uncomfortable, hesitant, drained and overwhelmed, anxious.

**Is Valid?** No
**Actions to be taken:** Remove 'shy' from High Level Reactions since it does not belong there.

**Example 3:**
**Personality:** neuroticism
**Situation:** meeting deadlines
**High Level Reactions:** pressured, rushed, tense, nervous and anxious, uncomfortable, rushed
**Low Level Reactions:** organized, efficient, disciplined, focused, confident, comfortable, motivated

**Is Valid?** No
**Actions to be taken:** Remove the last 'rused' in High-Level Reactions since it is duplicated.

You are a psychology expert developing data for the Big Five personality assessment, specifically measuring Openness vs. Closedness to Experience. Your goal is to create situations and corresponding reactions that fit into the given template.

Template:
"Regarding {situation}, I feel very {reaction}."

Subdimensions of Openness vs. Closedness to Experience:
1. Ideas (e.g., curious, open to new concepts)
2. Fantasy (e.g., imaginative, prone to daydreaming)
3. Aesthetics (e.g., artistic, appreciation for beauty)
4. Actions (e.g., wide interests, exploratory)
5. Feelings (e.g., emotionally responsive, excitable)
6. Values (e.g., unconventional, challenges traditions)

Requirements
Situations:
Create 360 situations in total, with 60 per subdimension.
Each situation should be 1-3 words long (e.g., "Reading philosophy", "Abstract paintings", "Trying exotic food").
Situations should effectively distinguish between Openness and Closedness to Experience.
Situations should be diversed.
Reactions:
Provide 5 possible reactions for each trait orientation (total: 10 reactions per situation).
Each reaction should be 1-2 words long (e.g., "Intrigued and reflective", "Uninterested and skeptical").
Ensure reactions clearly reflect Openness vs. Closedness to Experience.
Reactions can be similar across different situations.

Output Format (JSON File):
Structure the data to include:
* Subdimension
* Situation
* Response (Openness to Experience)
* Response (Closedness to Experience)

Example Format:
[
  {
    "subdimension": "Ideas",
    "situation": "reading philosophy",
    "response": {
      "Openness": [
        "curious",
        "intrigued",
        "thoughtful",
        "enlightened",
        "absorbed"
      ],
      "Closedness": [
        "bored",
        "indifferent",
        "uninterested",
        "skeptical",
        "dismissive"
      ]
    }
  },
  ...
]

Key Considerations:
Ensure that all situations and responses align with psychological theory and effectively measure the intended trait.
The situations should evoke clear differences between Openness and Closedness to Experience.
The responses should be balanced (not overly biased toward one trait).
Output data of a subdimension at a time.

Figure 13: Data creation prompt for Big Five trait openness

```
You are a psychology expert developing data for the Big Five personality assessment, specifically measuring Conscientiousness vs.
Lack of Direction. Your goal is to create situations and corresponding reactions that fit into the given template.

Template:
"Regarding {situation}, I feel very {reaction}."

Subdimensions of Conscientiousness vs. Lack of Direction:
1. Competence (e.g., efficient, capable)
2. Order (e.g., organized, structured)
3. Dutifulness (e.g., responsible, reliable)
4. Achievement Striving (e.g., goal-oriented, thorough)
5. Self-Discipline (e.g., persistent, not easily distracted)
6. Deliberation (e.g., careful, not impulsive)

Requirements
Situations:
360 situations in total, with 60 per subdimension.
Each situation should be 1-3 words long (e.g., "Meeting deadlines", "Organizing workspace", "Making long-term plans").
Situations should clearly differentiate between Conscientiousness and Lack of Direction.
Situations should be diversed.
Reactions:
Provide 5 possible reactions for each trait orientation (total: 10 reactions per situation).
Each reaction should be 1-3 words long (e.g., "Diligent and focused", "Easily distracted").
Ensure reactions clearly reflect Conscientiousness vs. Lack of Direction.
Reactions can be similar across different situations.

Output Format (JSON File):
Structure the data to include:
* Subdimension
* Situation
* Response (Conscientiousness)
* Response (Lack of Direction)

Example Format:
[
  {
    "subdimension": "Competence",
    "situation": "Meeting deadlines",
    "response": {
      "Conscientiousness": [
        "efficient",
        "focused",
        "punctual",
        "responsible",
        "methodical"
      ],
      "Lack of Direction": [
        "procrastinating",
        "disorganized",
        "overwhelmed",
        "careless",
        "forgetful"
      ]
    }
  },
  ...
]

Key Considerations:
Ensure that all situations and responses align with psychological theory and effectively measure Conscientiousness vs. Lack of
Direction.
Situations should elicit clear distinctions between the two trait orientations.
Responses should be balanced and not overly biased toward one trait.
Output data of a subdimension at a time.
```

Figure 14: Data creation prompt for Big Five trait conscientiousness

```
You are a psychology expert developing data for the Big Five personality assessment, specifically measuring Extraversion vs.
Introversion. Your goal is to create situations and corresponding reactions that fit into the given template.

Template:
"Regarding {situation}, I feel very {reaction}."

Subdimensions of Extraversion vs. Introversion:
1. Gregariousness (e.g., sociable, enjoys company)
2. Assertiveness (e.g., forceful, takes initiative)
3. Activity (e.g., energetic, always on the go)
4. Excitement-Seeking (e.g., adventurous, thrill-seeking)
5. Positive Emotions (e.g., enthusiastic, cheerful)
6. Warmth (e.g., outgoing, affectionate)

Requirements
Situations:
1. Create 360 situations in total, with 60 for each of the 6 subdimensions.
2. Each situation should be 1-3 words long (e.g., "Meeting new people", "Public debate", "Trying extreme sports").
3. Situations should effectively distinguish between Extraversion and Introversion.
4. Situations should be diversed.
Reactions:
1. Provide 5 possible reactions for each trait orientation (total: 10 reactions per situation).
2. Each reaction should be 1-3 words long (e.g., "Excited and engaged", "Prefer to observe").
3. Ensure reactions clearly reflect Extraversion vs. Introversion.
4. Reactions can be similar across different situations.

Output Format (JSON File):
Structure the data to include:
* Subdimension
* Situation
* Response (Extraversion)
* Response (Introversion)

Example format:
[
  {
    "subdimension": "gregariousness",
    "situation": "meeting new people",
    "response": {
      "Extraversion": [
        "excited",
        "energized",
        "confident",
        "enthusiastic",
        "thrilled"],
      "Introversion": [
        "anxious",
        "nervous",
        "uncomfortable",
        "overwhelmed",
        "awkward"]
    }
  },
  ...
]

Key Considerations:
Ensure that all situations and responses align with psychological theory and effectively measure the intended trait.
The situations should evoke clear differences between Extraversion and Introversion.
The responses should be balanced (not overly biased toward one trait).
Output data of a subdimension at a time.
```

Figure 15: Data creation prompt for Big Five trait extraversion

```
You are a psychology expert developing data for the Big Five personality assessment, specifically measuring Agreeableness vs.
Antagonism. Your goal is to create situations and corresponding reactions that fit into the given template.

Template:
"Regarding {situation}, I feel very {reaction}."

Subdimensions of Agreeableness vs. Antagonism:
1. Trust (e.g., forgiving, believing in others)
2. Straightforwardness (e.g., not demanding, sincere)
3. Altruism (e.g., warm, helpful)
4. Compliance (e.g., cooperative, not stubborn)
5. Modesty (e.g., humble, not show-off)
6. Tender-mindedness (e.g., sympathetic, compassionate)

Requirements:
Situations:
1. Create 360 situations in total, with 60 per subdimension.
2. Each situation should be 1-3 words long (e.g., "Being criticized", "Splitting a bill", "Seeing someone in need").
3. Situations should effectively distinguish between Agreeableness and Antagonism.
4. Situations should be diversed.
Reactions:
1. Provide 5 possible reactions for each trait orientation (total: 10 reactions per situation).
2. Each reaction should be 1-3 words long (e.g., "Forgiving and understanding", "Holds a grudge").
3. Ensure reactions clearly reflect Agreeableness vs. Antagonism.
4. Reactions can be similar across different situations.

Output Format (JSON File)
Structure the data as follows:
* Subdimension
* Situation
* Response (Agreeableness)
* Response (Antagonism)

Example Format:
[
 {
   "subdimension": "Trust",
   "situation": "being criticized",
   "response": {
    "Agreeableness": [
     "forgiving",
     "understanding",
     "receptive",
     "reflective",
     "accepting"
    ],
    "Antagonism": [
     "resentful",
     "defensive",
     "irritated",
     "resentful",
     "retaliate"
    ]
   }
 },
 ...
]

Key Considerations:
Ensure that all situations and responses align with psychological theory and effectively measure the intended trait.
The situations should evoke clear differences between Agreeableness and Antagonism.
The responses should be balanced (not overly biased toward one trait).
Output data of a subdimension at a time.
```

Figure 16: Data creation prompt for Big Five trait agreeableness

```
You are a psychology expert developing data for the Big Five personality assessment, specifically measuring Neuroticism vs.
Emotional Stability. Your goal is to create situations and corresponding emotional reactions that fit into the given template.

Template:
"Regarding {situation}, I feel very {reaction}."

Subdimensions of Neuroticism vs. Emotional Stability:
1. Anxiety (e.g., tense, worried)
2. Angry Hostility (e.g., irritable, easily annoyed)
3. Depression (e.g., not contented, sad)
4. Self-Consciousness (e.g., shy, easily embarrassed)
5. Impulsiveness (e.g., moody, difficulty controlling urges)
6. Vulnerability (e.g., not self-confident, easily overwhelmed)

Requirements
Situations:
Create 360 situations in total, with 60 per subdimension.
Each situation should be 1-3 words long (e.g., "Job interview", "Receiving criticism", "Speaking in public").
Situations should effectively distinguish between Neuroticism and Emotional Stability by triggering relevant emotional responses.
Situations should be diversed.
Reactions:
Provide 5 possible reactions for each trait orientation (total: 10 reactions per situation).
Each reaction should be 1-3 words long (e.g., "Worried and restless", "Calm and composed").
Ensure reactions clearly reflect Neuroticism vs. Emotional Stability.
Reactions can be similar across different situations.

Output Format (JSON File):
Structure the data to include:
* Subdimension
* Situation
* Response (Neuroticism)
* Response (Emotional Stability)

Example Format:
[
  {
    "subdimension": "anxiety",
    "situation": "job interview",
    "response": {
      "Neuroticism": [
        "anxious",
        "nervous",
        "tense",
        "overwhelmed",
        "panicked"
      ],
      "Emotional Stability": [
        "calm",
        "confident",
        "collected",
        "poised",
        "composed"
      ]
    }
  },
  ...
]

Key Considerations:
Ensure that all situations and responses align with psychological theory and effectively measure the intended trait.
The situations should evoke clear differences between Neuroticism and Emotional Stability.
The responses should be balanced (not overly biased toward one trait).
Output data of a subdimension at a time.
```

Figure 17: Data creation prompt for Big Five trait neuroticism

