# OpenReview forum: "From Traits to Circuits: Toward Mechanistic Interpretability of Personality in Large Language Models"
_ICLR.cc/2026/Conference — Submitted to ICLR 2026_

### Official Review · Reviewer_ZUG2 · 2025-10-21

**Soundness:** 3
**Presentation:** 2
**Contribution:** 3
**Rating:** 6
**Confidence:** 2

**Summary:**

This paper discusses personality in LLMs, pioneering the application of mechanistic interpretability to analyze the models themselves. The authors discovered that the identified circuits are functionally complete, structurally sparse, and heavily dependent on early MLP layers, which act as causal bottlenecks.

**Strengths:**

This is a thoroughly analyzed paper. While it does not introduce a novel methodology, it rigorously analyzes and investigates personality circuits. This approach uncovers many phenomena previously unknown to the community and refutes the conjecture that "personality is a globally diffuse property."

**Weaknesses:**

The paper's definition of personality is oversimplified. I do not believe the Big-Five model is sufficient to encapsulate personality, which could also include, for example, dark personality traits (e.g., the Dark Triad) or aspects such as values, beliefs, and motives. The "personality circuits" identified in this paper actually correspond to "Big-Five trait circuits," rather than the broader "personality circuits" as claimed. The authors should conduct further analysis on these aspects to reach a more definitive conclusion.

It is difficult for the paper to prove that these personality circuits are exclusive; they are very likely key components that are also involved in executing other semantic tasks.

The study was conducted on two relatively small and older LLMs. It remains unknown whether the conclusions can be generalized to state-of-the-art models, especially MoE-based architectures. An analysis of models like Qwen-3-30B-A3B and Qwen-3-235B-A22B would significantly enhance the paper's contribution.

**Questions:**

The template used to identify personality is highly structured. It is unclear whether similar phenomena persist in more realistic, user-focused tasks such as free-form conversation, extended dialogues, or long-form writing.

---

> ### Author Response · Authors · 2025-11-26
>
> > **W1. The paper's definition of personality is oversimplified.**
>
> We agree that the Big Five captures only part of human personality and excludes dimensions such as dark traits. In this work, we use the Big Five as a starting point, and we have clarified this scope in the revised version. Extending our method to broader personality constructs is a valuable direction that we plan to pursue in future work.
>
> > **W2. Personality circuits are not exclusive**
>
> Thank you for raising this point. We agree that the identified subgraphs are unlikely to be exclusive to personality-related behavior. Like many circuits in mechanistic interpretability, they may also participate in other semantic or contextual computations. Our goal is not to claim exclusivity, but to show that a sparse and consistent subset of components is sufficient to reproduce trait-aligned responses under controlled conditions.
>
> > **W3. Try MoE-based models**
>
> Thank you for the suggestion. We appreciate the reviewer’s point, and we will explore applying our method to larger and MoE-based models such as Qwen-3-30B-A3B and Qwen-3-235B-A22B in future work.
>
> > **Q1. The template is highly structured.**
>
> Thank you for the comment. We agree that the template is highly structured, this follows prior practice [1][2] in circuit analyses where the controlled inputs vary only in the targeted factors (personality level and situation in this paper). Extending the study to more realistic tasks is valuable. We plan to examine whether similar patterns persist in our future work.
>
>
> **References**
>
> [1] Knowledge Circuits in Pretrained Transformers
>
> [2] How Do LLMs Acquire New Knowledge? A Knowledge Circuits Perspective on Continual Pre-Training

---

> ### Comment · Reviewer_ZUG2 · 2025-11-26
>
> Thank you for your reply, but my concerns have hardly been addressed. I believe this paper may not yet be ready for publication and I would not object if others wished to reject it.

---

### Official Review · Reviewer_Z59u · 2025-10-23

**Soundness:** 1
**Presentation:** 2
**Contribution:** 2
**Rating:** 2
**Confidence:** 5

**Summary:**

This paper investigates whether the personality of large language models can be traced to identifiable transformer circuits. Using tools from the mechanistic interpretability community, the authors identify a small set of sparse nodes in a small, pretrained LLaMA model (LLaMA-2-Chat, 7B) that are responsible for generating answers on the proposed Trait-Trace dataset. Ablation studies and causal-intervention analyses show that certain nodes within these circuits can substantially influence LLM performance on the Trait-Trace task.

**Strengths:**

- The perspective of identifying interpretable circuits in Transformer models that are causally responsible for personality-like behaviors is interesting and could have important implications for safety, alignment, and the development of better chatbots.

- The paper is generally well written and easy to read.

**Weaknesses:**

- A major weakness lies in the evaluation. The study relies on a newly proposed Trait-Trace dataset generated by GPT-4o that focuses on single-word reactions to vignettes/trait prompts. All circuit-discovery and causal-intervention experiments depend on this fragile single-word reaction task. It is unclear what the task actually measures—the discovered circuits may merely capture distributional shifts in certain personality-related words rather than any higher-level notion of personality in LLMs. Generalization tests are essential. For example, under causal interventions/steering, do circuits discovered on Trait-Trace transfer to more complex settings (e.g., dialogue generation, storytelling, or psychometric evaluation items)? Given the authors’ access to trained psychology graduate students, such evaluations seem feasible. Demonstrating this would better justify the claim that the identified circuits reflect personality rather than confounding word-distribution shifts.

- The Trait-Trace task design is too simple. The template “I’m {p}, regarding {s}, I feel very {r}” biases lexical and affective choices, making the discovered circuits specific to particular word choices rather than to general personality constructs. It remains unclear what construct this task is evaluating.

- Limited conceptual insight. As framed, one could likely find circuits or sparse subgraphs for almost any language-model behavior. The authors should better demonstrate—or at least discuss—why these circuits matter and why the discovered early-layer MLP features align with human intuitions.

**Questions:**

If you replace the prompt with random fillers unrelated to personality, would the intervened circuits still induce a similar shift in the output-logit distribution?

---

> ### Author Response · Authors · 2025-11-26
>
> > **W1: Limited evaluation**
>
> We appreciate the reviewer’s concern. In this work, we focus on trait-aligned behavior under controlled prompts, and do not assume that the model has real personality. The reactions in TraitTrace are short phrases or clauses (often multi-token). We see this controlled setup as a first step that makes circuit analysis tractable, and we agree it will be important in future work to test whether the discovered circuits transfer to richer settings such as dialogue, storytelling, or standardized personality assessments.
>
> > **W2: Template is too simple**
>
> Thank you for your comment. Our goal is to see whether the model shows trait-aligned behavior across different situations, rather than to model personality in full. To keep the experiment controlled, we use the fixed template “I’m {p}, regarding {s}, I feel very {r}” so that syntax and framing stay the same, and the main variation comes from the personality description and the situation.
>
> > **W3: Limited conceptual insight.**
>
> We appreciate the reviewer’s perspective. The importance of these circuits lies in the fact that prior work on LLM “personality” has been almost entirely prompt- or questionnaire-based, treating it as a black-box phenomenon. Our work moves beyond prompts and shows that trait-aligned behavior is supported by identifiable internal circuits. In the revised version, we add a brief discussion of why these personality circuits matter and possible explanations for the prominent role of early-layer MLP features.
>
> > **Q1: Random-filler prompt control for circuit interventions**
>
> Thank you for the insightful question. We additionally conduct an experiment using the Openness-high and Openness-low circuits. For each circuit, we randomly sample 10 data points from TraitTrace and compare two settings: (1) the original template “I’m {p}, regarding {s}, I feel very {r}”, and (2) a version where the prompts are replaced with length-matched random tokens unrelated to personality. Under the original template, the output logits shift toward trait-aligned tokens. Under the random-token prompts, no clear pattern emerges and the logits over trait-related tokens are close to random. This suggests that these circuits are context-dependent computations that are recruited only when the relevant personality description is present. We will include a discussion of this experiment in the next revision of the paper.

---

### Official Review · Reviewer_zxtA · 2025-10-27

**Soundness:** 1
**Presentation:** 2
**Contribution:** 1
**Rating:** 2
**Confidence:** 4

**Summary:**

This paper explores whether personality traits (based on the Big Five model) can be localized as identifiable “circuits” within large language models (LLMs). The authors construct a synthetic dataset, TRAITTRACE, containing prompts that express high or low levels of each trait and corresponding trait-consistent reactions. Using Edge Attribution Patching with Integrated Gradients (EAP-IG), they identify minimal subgraphs within the model that preserve performance on trait-consistent response prediction. Results suggest that small, sparse circuits can reproduce the full model’s behavior, that high and low levels of traits share many nodes but differ in edge directions, and that early MLP layers serve as bottlenecks for trait information.

**Strengths:**

This paper tries to move beyond behavioral probing toward the mechanistic interpretability of social/psychological constructs.

**Weaknesses:**

## Weaknesses and Suggestions

### 1. The motivation is weak.
The authors justify this work through an analogy with neuroscience, arguing that personality traits in humans arise from neural circuits and therefore may also emerge as “trait circuits” in LLMs. However, this analogy is conceptually flawed. Human traits are latent psychological dimensions, not localized neural entities, and the connection to artificial circuits is purely metaphorical.

---

### 2. Ethical statement is missing.
Because the paper draws direct analogies between human brain circuits and model activations, it risks **anthropomorphism**, suggesting that LLMs “possess” personality traits or human-like psychology. Such framing requires careful ethical consideration and a clear disclaimer, but no ethical statement is provided. The authors should explicitly acknowledge these limitations and clarify that their findings do not imply genuine human-like cognition.

---

### 3. Experimental rigor is low.
Only two small instruction-tuned models (LLaMA-2-7B-Chat and Phi-2) are tested, without including base or larger models. This makes it difficult to assess whether the findings generalize across training phases or scales of LLMs. Including additional models or verifying whether similar trait circuits emerge in non-chat variants would significantly strengthen the paper. Also, I understand that this paper's goal is to discover the circuits lying in the LLMs. But to evaluate the quality and validity of the dataset, I recommend that authors provide the evaluation on other methods, such as pure prompting.

---

### 4. Prompt and task design are conceptually flawed.
The Big Five traits are continuous spectra, but the dataset reduces them to binary self-descriptions (e.g., “I am high in openness” vs. “I am low in openness”). This introduces strong lexical cues and risks capturing superficial associations between trait names and responses rather than genuine trait inference. A more realistic approach would involve inferring traits from open-ended essays or autobiographical texts. This method is one of the canonical methods to evaluate the personalities of humans [1].
For methodological reference, see [2].

---

### 5. Novelty is limited.
The technical approach, combining edge-attribution patching with pruning, is a direct application of existing interpretability methods. The main novelty lies in dataset curation, but the dataset curation is not rigorous enough.

---

### 6. Causal interpretation is overstated.
In Section 6.3, the authors conduct causal intervention analysis. However, the key question, whether these circuits truly represent personality traits rather than lexical correlations, remains unresolved. Without ruling out such confounders, it is premature to claim that the identified subgraphs mechanistically encode traits.

[1] McAdams, Dan P. "Narrative identity." Handbook of identity theory and research. New York, NY: Springer New York, 2011. 99-115.

[2] Suh, Joseph, et al. "Rediscovering the latent dimensions of personality with large language models as trait descriptors." arXiv preprint arXiv:2409.09905 (2024).

**Questions:**

1. **Evaluation details.** The details of the evaluation are not fully provided. The words or phrases like "procrastinating" are divided into 5 tokens. And the LLM response can be like a sentence, but how authors check the overlap between the references and LLM-generated tokens is not fully explained.

2. **Dataset details.** Please provide more details of the curated dataset to evaluate the quality.

---

> ### Author Response · Authors · 2025-11-26
>
> > **W1: The motivation is weak.**
>
> Thank you for the comment. We agree that human personality traits are psychological constructs rather than discrete neural modules. Our reference to neuroscience was intended only as a motivation for the question we ask, not as a one-to-one analogy. Building on findings that different Big Five traits correlate with distinct patterns of neural connectivity in the brain, we ask whether LLMs might likewise rely on sparse and reproducible subgraphs when expressing trait-aligned behavior.
>
> > **W2: Ethical statement is missing.**
>
> Thank you for raising this point. We have added an Ethics and Limitations section that makes explicit that our work does not attribute human-like personality, cognition, or mental states to LLMs.
>
> > **W3: Experimental rigor is low.**
>
> We thank the reviewer for raising this important point. We have now extended our analysis to LLaMA-2-7B-Base, the base version of LLaMA-2-7B-Chat without instruction tuning, to check whether the circuit patterns we observe also hold beyond instruction-tuned models.
>
> The new results, shown in Tables 1–6 below, aligned with what we see in LLaMA-2-7B-Chat and Phi-2:
> - Circuit compactness and fidelity (see Table 1).
>   Circuits extracted from LLaMA-2-7B-Base keep only about 8% of nodes and 0.04% of edges, yet still reach a Hit@10 of 0.96 on both the analysis and test sets, essentially matching the full model’s 0.97.
> - Intra-trait analysis (see Table 2).
>   High overlap between high and low levels of each trait (e.g., 0.83 node overlap for Extraversion) again suggests that each trait dimension relies on a shared structural scaffold, as in LLaMA-2-7B-Chat and Phi-2.
> - inter-trait analysis (see Tables 3–5).
>   Neuroticism remains the most structurally distinct, with the lowest overlap with other traits, echoing the pattern we observe in the other models.
> - Causal intervention (Table 6).
>   Early MLP layers show up as key bottlenecks in LLaMA-2-7B-Base, LLaMA-2-7B-Chat, and Phi-2. In LLaMA-2-7B-Base, for example, ablating m1 leads to a large 0.96 drop in performance, in line with our earlier findings.
>
> Taken together, these results suggest that the patterns we report are stable across architectures, training phases, and model sizes. We will add these supplementary results to the revised version to make this point clearer.
>
> Regarding dataset evaluation, the “Model” scores in Table 1 in the paper reflect the performance of the full model using the prompt template described in Section 4.1. This serves as a pure-prompting baseline showing how well the complete model aligns with the trait-consistent reactions. The “Circuit” scores are obtained using the same prompts but limiting computation to the sparse subgraph identified by our method. This setup allows us to examine how much of the full-model behavior can be reproduced by a small subset of components. We have revised the manuscript to make this setup clearer.
>
> > **W4: Prompt and task design are conceptually flawed.**
>
> Thank you for the comment. We agree that Big Five traits are continuous. In this paper, we aim to identify the internal computation that supports trait-conditioned generation, which requires controlled inputs that vary only in the targeted trait factor. Using binary self-descriptions is a practical way to isolate this variable and avoid confounds that would make circuit attribution unreliable. This design does not assume personality is discrete, and our reactions span diverse lexical forms far beyond the trait labels. We see the extension of circuit analysis to more naturalistic, open-ended contexts as an important avenue for future work.
>
> > **W5: Novelty is limited.**
>
> Thank you for the comment. Our contribution is not a new attribution algorithm, but a new problem framing and the first mechanistic analysis of personality expression in LLMs. The main contributions of the paper are threefold:
> 1. We frame personality in LLMs as a mechanistically interpretable problem and introduce TRAITTRACE, a controlled dataset that enables circuit-level analysis rather than black-box probing.
> 2. We identify trait-specific circuits and validate them through ablation, layer-wise structure, and trait-overlap analysis.
> 3. We conduct causal interventions showing that a small number of early-layer components exert targeted influence on personality expression.
> We believe these points reflect the novelty of the work.
>
> > **W6: Causal interpretation is overstated.**
>
> Thank you for the comment. To clarify, the causal analysis in Section 6.3 is meant to assess how individual nodes within the identified circuits affect trait-aligned behavior. It is not intended to suggest that these circuits correspond to personality traits. We have adjusted the wording in the revised text to make this scope clearer.

---

> > ### Author Response · Authors · 2025-11-26
> > **Supplementary Results on LLaMA-2-7B-Base**
> >
> > Table 1 Hit@10 scores for the full model (G) and the standalone circuit (C) on the analysis and test sets of TraitTrace.
> >
> > | Big Five Traits   | Level | Node %  | Edge % | 𝒟_analysis | 𝒟_analysis | 𝒟_test | 𝒟_test |
> > |-------------------|-------|---------|--------|------------|------------|--------|--------|
> > |                   |       |         |        | Model (𝒢)  | Circuit (𝒞) | Model (𝒢) | Circuit (𝒞) |
> > | Openness          | High  | 5.87%   | 0.03%  | 0.94       | 0.93       | 0.94   | 0.93   |
> > |                   | Low   | 7.15%   | 0.03%  | 0.98       | 0.98       | 0.98   | 0.98   |
> > | Conscientiousness | High  | 9.04%   | 0.04%  | 0.99       | 0.92       | 0.99   | 0.92   |
> > |                   | Low   | 11.43%  | 0.05%  | 0.93       | 0.93       | 0.93   | 0.94   |
> > | Extraversion      | High  | 9.33%   | 0.04%  | 0.98       | 0.98       | 0.98   | 0.98   |
> > |                   | Low   | 8.25%   | 0.04%  | 0.95       | 0.95       | 0.95   | 0.95   |
> > | Agreeableness     | High  | 6.85%   | 0.04%  | 1.00       | 1.00       | 1.00   | 1.00   |
> > |                   | Low   | 6.06%   | 0.03%  | 0.98       | 0.97       | 0.98   | 0.97   |
> > | Neuroticism       | High  | 9.11%   | 0.04%  | 0.95       | 0.94       | 0.95   | 0.94   |
> > |                   | Low   | 7.23%   | 0.03%  | 0.98       | 0.98       | 0.98   | 0.98   |
> > | Average       | -     | 8.03%   | 0.04%  | 0.97       | 0.96       | 0.97   | 0.96   |
> >
> > Table 2 Intra-trait circuit overlap between high and low levels of each personality trait
> >
> > | Trait             | Node Overlap | Edge Overlap |
> > |-------------------|-------------:|-------------:|
> > | Openness          | 0.77         | 0.63         |
> > | Conscientiousness | 0.74         | 0.87         |
> > | Extraversion      | 0.83         | 0.80         |
> > | Agreeableness     | 0.78         | 0.75         |
> > | Neuroticism       | 0.63         | 0.61         |
> >
> >
> > Table 3 Average inter-trait circuit overlap
> >
> > | Trait             | Node Overlap | Edge Overlap |
> > |-------------------|-------------:|-------------:|
> > | Openness          | 0.53         | 0.25         |
> > | Conscientiousness | 0.55         | 0.23         |
> > | Extraversion      | 0.51         | 0.17         |
> > | Agreeableness     | 0.52         | 0.21         |
> > | Neuroticism       | 0.42         | 0.12         |
> >
> >
> > Table 4 Inter-trait circuit overlap (Node Overlap)
> > |               | Openness | Conscientiousness | Extraversion | Agreeableness | Neuroticism |
> > |--------------|----------|-------------------|--------------|---------------|-------------|
> > | Openness          | 1.00 | 0.60 | 0.49 | 0.54 | 0.48 |
> > | Conscientiousness | 0.60 | 1.00 | 0.61 | 0.58 | 0.40 |
> > | Extraversion      | 0.49 | 0.61 | 1.00 | 0.55 | 0.37 |
> > | Agreeableness     | 0.54 | 0.58 | 0.55 | 1.00 | 0.42 |
> > | Neuroticism       | 0.48 | 0.40 | 0.37 | 0.42 | 1.00 |
> >
> > Table 5 Inter-trait circuit overlap (Edge Overlap)
> > |                      | Openness | Conscientiousness | Extraversion | Agreeableness | Neuroticism |
> > |----------------------|----------|-------------------|--------------|---------------|-------------|
> > | Openness          | 1.00 | 0.33 | 0.19 | 0.32 | 0.17 |
> > | Conscientiousness | 0.33 | 1.00 | 0.25 | 0.22 | 0.11 |
> > | Extraversion      | 0.19 | 0.25 | 1.00 | 0.21 | 0.04 |
> > | Agreeableness     | 0.32 | 0.22 | 0.21 | 1.00 | 0.07 |
> > | Neuroticism       | 0.19 | 0.13 | 0.06 | 0.09 | 1.00 |
> >
> > Table 6 Top 5 nodes ranked by mean accuracy drop
> > | Node   | Mean Drop |
> > |--------|-----------|
> > | m1     | 0.96      |
> > | m0     | 0.15      |
> > | a9.h1 | 0.08      |
> > | m31 | 0.05      |
> > | m8    | 0.04      |

---

> ### Author Response · Authors · 2025-11-26
>
> > **Q1: Evaluation details.**
>
> Thank you for pointing this out. In our evaluation, we use a prefix-matching scheme. During evaluation, we check whether the model’s generated output begins with any reference reaction. if so, we count it as a hit. We have revised the text for better clarity.
>
> > **Q2: Dataset details.**
>
> Thank you for your question. The details of the human evaluation for TraitTrace can be found in Section 11. Four graduate students with a background in psychology reviewed a random sample of 200 entries, and 93.5% were judged as valid.

---

### Official Review · Reviewer_g9iQ · 2025-11-01

**Soundness:** 3
**Presentation:** 3
**Contribution:** 4
**Rating:** 8
**Confidence:** 4

**Summary:**

the paper studies whether personality in LLMs may similarly be realized through structured internal computation paths.

The authors come to the conclusion that "only a compact set of attention heads and MLP units appears necessary
for encoding and expressing each trait across different models".

**Strengths:**

The research is on a very timely and interesting topic. The results are convincing and should be interesting for a broad community of researchers.

**Weaknesses:**

Though authors correctly mention that LLMs simulate certain behaviour or personality they do antropomorphize LLMs in other parts of the text. This is unfortunate but minor problem in my opinion.

**Questions:**

How general is your approach and would it be applicable to bigger models? In particular how can we be sure that the conclusions will hold in the models that are order of magnitude bigger and go through a significantly longer post-training phase for alignment?

---

> ### Author Response · Authors · 2025-11-26
>
> > **W1: Anthropomorphizing LLMs**
>
> Thank you for raising this point. We fully agree that LLMs simulate personality-like behaviors rather than possessing genuine personality traits. We will revise the wording in the next version to avoid anthropomorphizing LLMs when describing the models.
>
> > **Q1: Generalization to larger and more heavily aligned models**
>
> Thank you for the thoughtful question. Our approach is applicable to all GPT-style models, and it scales well because each analysis requires only two forward passes and one backward pass per input. In this paper, we observe consistent patterns across LLaMA-2-7B-Chat and Phi-2, despite their differences in size and whether they have undergone RLHF. We appreciate the reviewer’s interest in the broader generalization of our findings, and we plan to explore larger models in future work.

---

### Meta-Review · Area_Chair_NV5E · 2026-01-03

**Summary:**

The paper explores whether personality-like behaviors in LLMs can be localized to sparse “trait circuits” using a new dataset (TraitTrace) and attribution patching techniques.

Reviewers agree the topic is timely and the analysis is thorough, but they consistently question the conceptual grounding, dataset validity, and scope of conclusions. The work relies heavily on synthetic, template-based inputs that oversimplify personality constructs and risk capturing lexical correlations rather than meaningful mechanistic structure. Evaluation is limited to small, older models and controlled prompts, leaving unclear whether findings generalize to realistic tasks, larger architectures, or more naturalistic personality expressions. Overall, despite an interesting direction, the paper lacks the rigor and conceptual clarity needed for acceptance.

**Reviewer Concerns:**

The rebuttal adequately clarifies several minor issues (anthropomorphizing language, evaluation details, and ethics statement) and adds supplementary results on LLaMA-2-Base. However, core concerns remain unresolved: the conceptual motivation is weak, the TraitTrace dataset is too narrow and biased toward lexical shortcuts, and the evaluation task is too simplistic to support claims about “personality circuits.” Reviewers also note that the findings may reflect shallow distributional features rather than meaningful psychological constructs, and the causal claims remain insufficiently supported. Limited generalization across models, architectures, and realistic prompt settings further undermines the strength of the conclusions. These fundamental issues justify a recommendation to reject.

**Reviewer Scores:**

Reviewer g9iQ: Likely unchanged (8), as their evaluation was already positive and concerns were minor.

Reviewer zxtA: Likely unchanged (2), as key objections regarding motivation, dataset design, and conceptual validity remain.

Reviewer Z59u: Likely unchanged (2), since major concerns about construct validity and generalization were not addressed.

Reviewer ZUG2: Likely unchanged or slightly lowered (6 to 4), given their final comment that concerns remain largely unresolved.

---

### Decision · Program_Chairs · 2026-01-26

Reject